# The genomic epidemiology of *Escherichia albertii* infecting humans and birds in Great Britain

Rebecca J. Bengtsson[1], Kate S. Baker [1]✉, Andrew A. Cunningham [2], David R. Greig[3], Shinto K. John[2], Shaheed K. Macgregor[4], Katharina Seilern-Moy[2], Simon Spiro[4], Charlotte C. Chong[1], P Malaka De Silva[1], Claire Jenkins[3] & Becki Lawson[2]

*Escherichia albertii* is a recently identified gastrointestinal bacterial pathogen of humans and animals which is typically misidentified as pathotypes of diarrhoeagenic *Escherichia coli* or *Shigella* species and is generally only detected during genomic surveillance of other Enterobacteriaceae. The incidence of *E. albertii* is likely underestimated, and its epidemiology and clinical relevance are poorly characterised. Here, we whole genome sequenced *E. albertii* isolates from humans ($n = 83$) and birds ($n = 79$) isolated in Great Britain between 2000 and 2021 and analysed these alongside a broader public dataset ($n = 475$) to address these gaps. We found human and avian isolates typically (90%; 148/164) belonged to host-associated monophyletic groups with distinct virulence and antimicrobial resistance profiles. Overlaid patient epidemiological data suggested that human infection was likely related to travel and possibly foodborne transmission. The Shiga toxin encoding *stx2f* gene was associated with clinical disease (OR = 10.27, 95% CI = 2.98–35.45 $p = 0.0002$) in finches. Our results suggest that improved future surveillance will further elucidate disease ecology and public and animal health risks associated with *E. albertii*.

*Escherichia albertii*, a Gram-negative gastrointestinal pathogen of humans and animals, was first confirmed as a novel bacterium in 2003[1–3]. This pathogen is often misidentified because it is difficult to differentiate from *Shigella* species as they are morphologically, colonially, metabolically and biochemically similar; for example, both are non-lactose fermenting and lysine decarboxylase negative[4,5]. Implementation of PCR for the detection of a wide range of gastrointestinal (GI) pathogens, including diarrhoeagenic *E. coli* (DEC), and the use of whole genome sequencing (WGS) for identification and typing has provided a more robust and reliable approach for the detection and characterisation of *E. albertii*[6–8]. As a result of the implementation of routine WGS for microbiological surveillance by the United Kingdom

Health Security Agency (UKHSA, formerly Public Health England), we now have the capacity to accurately identify *E. albertii* in individuals presenting to primary healthcare settings with gastrointestinal symptoms.

Although detection and speciation prior to the genomic era were challenging, the pathogenic traits of *E. albertii* are well described[4]. Like certain DEC pathotypes, specifically the enteropathogenic *E. coli* (EPEC) and a subset of Shiga toxin-producing *E. coli* (STEC), the genome of *E. albertii* contains the locus of enterocyte effacement (LEE) pathogenicity island encoding a type III secretion system involved in the attachment of the pathogen to the gut mucosa[9,10]. Colonisation of EPEC and *eae* gene positive (a marker of LEE) STEC in both humans and

[1]Clinical Infection, Microbiology and Immunology, Institute of Infection, Veterinary and Ecological Sciences, The University of Liverpool, Liverpool, UK. [2]Institute of Zoology, Zoological Society of London, Regent's Park, London NW1 4RY, UK. [3]Gastrointestinal and Food Safety (One Health) Division, UK Health Security Agency, Colindale, London, UK. [4]Wildlife Health Services, Zoological Society of London, Regent's Park, London NW1 4RY, UK. ✉e-mail: kbaker@liverpool.ac.uk

animals can lead to the formation of attaching and effacing (A/E) lesions on the intestinal epithelial cells[11]. Cytolethal distending toxin (cdt) is encoded by the *cdtABC* operon and is classified into five sub-types based on sequence variation of the *cdtB* gene (*cdtB*-I to *cdtB*-V). Of these, *cdtB* subtypes I/II/III/V have been identified in *E. albertii*[10,12]. The *stx* gene encoding for Shiga toxins, predominantly the *stx2f* sub-type, has been found in certain strains of *E. albertii*[13]. Although these virulence determinants are well described in *E. albertii*, their distribution and clinical relevance across species require further elucidation.

Clinical symptoms in human patients caused by *E. albertii* infection are similar to those caused by EPEC and typically include watery diarrhoea, dehydration, abdominal pain, vomiting and fever[13,14]. Over the last decade, outbreaks of GI disease in people in Japan have been attributed to *E. albertii* following re-examination of the original microbiological findings using genomic typing methods, such as multilocus sequence typing (MLST)[15,16]. However, because of the challenges around detection and identification, which have hampered systematic surveillance, data on the epidemiology, source and transmission routes of *E. albertii* infections are sparse. In England to date, commercial GI PCR panels have been adopted by approximately 25% of diagnostic microbiology laboratories in the National Health Service network[17]. Furthermore, not all the commercial GI PCR panels target *eae*, and not all diagnostic laboratories refer samples to the Gastrointestinal Bacteria Reference Unit (GBRU) at UKHSA for further identification. These limitations of the current surveillance mechanisms likely result in a considerable under-ascertainment of cases, and the true burden of human infection caused by *E. albertii* remains unknown.

In addition to infecting people, *E. albertii* can infect birds and other animals, in which the prevalence and pathogenicity are also unclear. In the mid-1990s, multiple mortality incidents of Fringillidae (finch) species were observed in Scotland with a bacterium, later identified as *E. albertii*, hypothesised to be the cause of death[2,18,19]. Similarly, in 2004, large-scale mortality of a finch species (*Carduelis flammea*) occurred in Alaska, United States of America (USA), with *E. albertii* as the probable aetiology[2]. Active molecular surveillance studies for *E. albertii* have since detected the bacterium in dead and apparently healthy birds of multiple orders and species from Australia, Asia, mainland Europe and North America[2,20–23]. *Escherichia albertii* also has been detected in poultry faeces/GI tract contents and meat[8,24–26] and in domestic mammals (e.g., pig, cat) and both terrestrial and marine wild mammal species (e.g., raccoon, seal, bat)[8,27,28]. Although the occurrence and significance to mammal host health remain uncertain, there is a growing body of evidence that avian hosts may act as a reservoir of infection[21,26]. Thus, the extent of associated

diseases in birds and the relationship of bird and human infections requires further investigation.

Here, we performed WGS analysis on *E. albertii* isolates from humans and birds in Great Britain (GB) from archives held at the UKHSA and the Zoological Society of London (ZSL), respectively, to investigate the epidemiology of this recently identified pathogen. The aims of the study were to integrate the phylogenetic and epidemiological data in order to gain insights into the ecology of *E. albertii* among people and birds, to better understand the risk factors (e.g., recent international travel) associated with human infection, and to infer the likely significance of *E. albertii* infection to avian host health. Owing to the relative importance of *Enterobacteriaceae* as a reservoir for antimicrobial resistance (AMR) genes, we also describe and compare genotypic AMR profiles recovered from the two host groups.

## Results

### Summary of the human isolates
Between January 2014 (when routine WGS was first implemented at the GBRU) and December 2021, 83 isolates from human cases were confirmed as *E. albertii*. Over this 8-year period, between 4 and 23 isolates were identified per year (Supplementary Fig. 1). Metadata regarding patient gender, age and history of recent travel was available for 82, 83 and 26 isolates, respectively (Supplementary Data 1). There was no statistical association of isolates with gender (39 males, 32 females) and little association with age group (Table 1, Fig. 1, Supplementary Fig. 1). A total of 24 (29%, $n = 24/83$) patients stated they had recently travelled (within 7 days of onset of symptoms) outside the UK, of which the majority ($n = 21/24$, 88%) reported travel to Asia. Travel status was unknown for the remaining cases (71%, $n = 59/83$), as their travel history was not recorded.

### Summary of the bird isolates
Seventy-four *E. albertii* isolates from wild birds were analysed over the period 2000–2019 inclusive. With a single exception (tawny owl *Strix aluco*), the hosts were Passeriformes from the following families in declining rank order: Fringillidae $n = 50$, Passeridae $n = 8$, Turdidae $n = 7$, Paridae $n = 4$ and single birds from the Hirundinidae, Motacillidae, Prunellidae and Sturnidae (for species composition see Supplementary Data 2). Isolates were identified each year across the 20-year study period with two exceptions and from a total of 72 sites. Available data permitted the determination of the inferred significance of *E. albertii* to host health (see Supplementary methods) for 69 wild birds, with 38% ($n = 26$) being significant, 46% ($n = 32$) being equivocal and 16% ($n = 11$) being incidental. The wild birds for which *E. albertii*

**Table 1 | Phylogenetic and epidemiological features of *Escherichia albertii* Bayesian Analysis of Population Structure (BAPs) clusters for isolates from Great Britain**

| BAPS cluster | Genomic features | | Isolate composition | | | | Statistical support and nomenclature | |
| | Congruence with phylogeny | Average pair-wise distance | Total iso-lates (n) | Human (% of cluster) | Wild bird (% of cluster) | Captive bird (% of cluster) | Proportion humans [Proportion (95% confidence interval), two-tailed p-value] | Final determination[a] |
|---|---|---|---|---|---|---|---|---|
| 1 | Monophyletic | 79 | 4 | 4 (100) | 0 (0) | 0 (0) | 1.00 (0.40–1.00), $p = 0.1232$ | HAC |
| 2 | Monophyletic | 28 | 17 | 17 (100) | 0 (0) | 0 (0) | 1.00 (0.80–1.00), $p < 0.0001$ | HAC |
| 3 | Monophyletic | 132 | 7 | 7 (100) | 0 (0) | 0 (0) | 1.00 (0.59–1.00), $p = 0.0156$ | HAC |
| 4 | Monophyletic | 25 | 16 | 16 (100) | 0 (0) | 0 (0) | 1.00 (0.79–1.00), $p < 0.0001$ | HAC |
| 5 | Monophyletic | 167 | 4 | 4 (100) | 0 (0) | 0 (0) | 1.00 (0.40–1.00), $p = 0.1232$ | HAC |
| 6 | Monophyletic | 967 | 61 | 12 (20) | 49 (80) | 0 (0) | 0.20 (0.11–0.33), $p < 0.0001$ | BAC |
| 7 | Monophyletic | 73 | 29 | 4 (14) | 24 (86) | 1 (3) | 0.17 (0.05–0.35), $p = 0.0003$ | BAC |
| 8 | Polyphyletic | 3169 | 24 | 19 (79) | 1 (4) | 4 (17) | 0.80 (0.58–0.93), $p = 0.0148$ | HAC |
| Total | NA | NA | 162 | 85 | 74 | 5 | | |

[a]BAC = Bird-associated cluster, HAC = human-associated cluster.

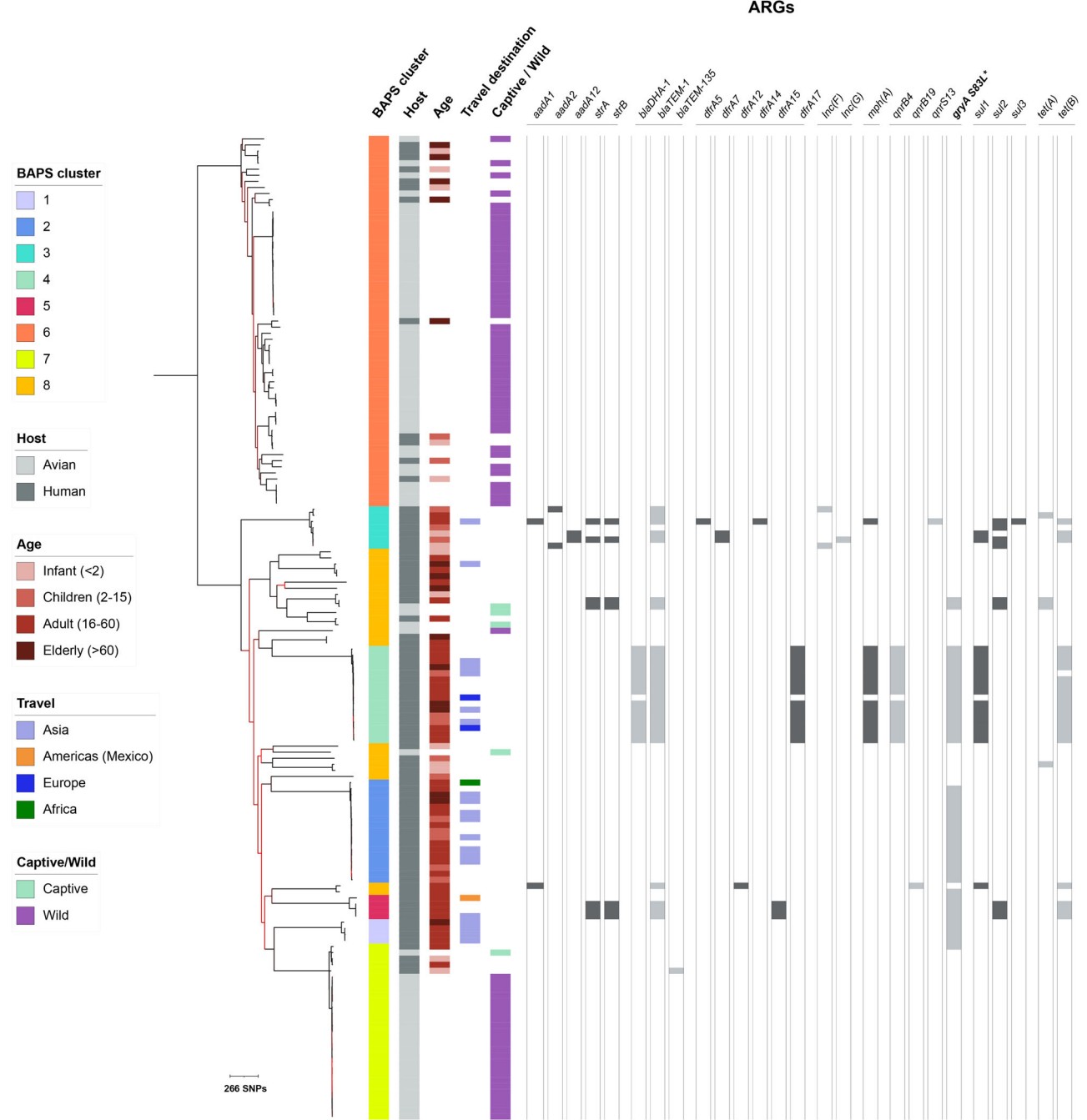

**Fig. 1 | Midpoint rooted maximum likelihood phylogenetic tree of *Escherichia albertii* isolates from Great Britain showing human demographic features and antimicrobial resistance genes (ARGs).** The scale bar is shown in single nucleotide polymorphisms (SNPs). Isolate metadata are displayed in the adjacent tracks on the right according to the inlaid keys on the left (BAPS = Bayesian Analysis of Population Structure). Tracks in the centre panel show the presence of ARGs grouped by antimicrobial class, with the *gryA* S83L point mutation highlighted in bold and indicated with an asterisk. Phylogenetic branches highlighted in red indicate nodes with low bootstrap support (between 50 and 70%).

infection was considered significant to host health comprised Fringillidae (bullfinch *Pyrrula pyrrhula* n = 1, chaffinch *Fringilla coelebs* n = 4, greenfinch *Chloris chloris* n = 9 and siskin *Spinus spinus* n = 8), house sparrow *Passer domesticus* n = 3 and a single blue tit *Cyanistes caeruleus*.

The five isolates from captive zoo birds were from a diverse range of species (Anseriformes, Passeriformes, Pelecaniformes and Spheniscilformes). Inferred significance to host health was categorised as significant for one captive bird (black-footed penguin *Spheniscus demursus*), equivocal for two cases and incidental for two cases.

Fringillidae was more frequently associated with 'significant' inferred clinical significance than non-Fringillidae species combined (p = 0.0433, Fisher's exact test) for the wild and captive bird data, and this was also well supported statistically among wild birds alone (p = 0.0612).

## Genomic epidemiology of *Escherichia albertii* from humans and bird isolates

To explore the genomic epidemiology of *E. albertii* among the human and bird isolates from GB, demographic features were overlaid on the

bacterial population structure and statistical support for associations with metadata variables was evaluated.

Specifically, to determine the population structure of *E. albertii* within our dataset, a maximum likelihood phylogeny was constructed based on an SNP alignment of 26,594 bp (Fig. 1). BAPS identified eight clusters consistent with monophyletic clustering, with the exception of BAPS cluster 8, which was split across multiple regions of the tree (Table 1, Fig. 1). Combining the epidemiological information with this population structure revealed distinct and separate phylogenetic clustering of bird and human isolates ($p < 0.0001$, Chi-square test, 7 df), although statistical support varied for individual clusters (see Table 1). Most bird isolates ($n = 74/79$) belonged to BAPS clusters 6 and 7, in which bird isolates were statistically over-represented, and these were termed bird-associated clusters (BACs, Table 1). To facilitate further high-level investigation, BAPS clusters 1, 2, 3, 4, 5 and 8 were termed human-associated clusters (HACs). Intermixing between human and bird isolates was observed within both BACs and one HAC. Specifically, the HAC BAPS 8 contained 6% ($n = 5/79$) of isolates from birds, 4/5 of which were from captive zoo birds. Within the BACs 6 and 7, 18% ($n = 16/90$) of isolates were from humans.

To investigate the association of *E. albertii* with human demographic features, we associated travel history and patient age with the bacterial population structure. All 24 isolates from human patients with a confirmed recent history of international travel belonged to HACs, and at least one travel-associated isolate was identified in each of the six HACs (Fig. 1). The travel status was not recorded for any of the human cases with isolates that fell within the BACs. When associating human age groups with population cluster assignation (BAC/ HAC), we observed a significant difference between the BACs and HACs ($p = 0.0008$, Fisher's exact, Fig. 1, Supplementary Data 3). Within the BACs, infants (<2 years) and older people (>60 years) were the predominant human age groups, comprising 44% ($n = 7/16$) and 31% ($n = 5/16$) of human isolates, respectively (where patient age information was available, Supplementary Data 1). In contrast, the predominant age group within the HACs was adult (16–60 years), comprising 55% ($n = 37/67$) of human isolates.

### Virulence profiles and associations with disease in bird hosts

The *eae* gene was present in all but one isolate within the dataset, and the *cdtA*, *cdtB*, and *cdtC* genes were present in >94% ($n = 153/162$) isolates (Fig. 2, Supplementary Fig. 2). The *stx2f* gene was detected in 38 isolates, the majority ($n = 37/38$, 97%) of which were from wild birds in BACs, except for one human isolate (SRR6144114) belonging in BAPS 8. Among the wild birds, *stx2f* resulted in an increased odds of inferred clinical significance of infection (relative to equivocal and incidental combined) (OR = 10.27, 95% CI = 2.98–35.45, $p = 0.0002$). There was little evidence for confounding of the disease association by bird family (Fringillidae/Non-Fringillidae, adjusted OR = 10.25, 95% CI = 2.66–92.78). A possible effect modification of the bird family (Strata specific OR: OR = 12.68, 95% CI = 2.66–877.38, *p*-value < 0.001 (Fringillidae), OR = 0.64, 95% CI = 0.03–16.03, *p*-value = 1) was challenging to evaluate further as the *stx2f* was over-represented among the Fringillidae (vs non-Fringillidae OR = 25.67, 95% CI = 5.35–123.23, $p = 0.0001$), specifically of 37 *stx2f*-positive bird isolates, 35 were from Fringillidae species.

### Antimicrobial resistance profiles in human and bird isolates

To investigate the genotypic predictors of AMR among *E. albertii* isolates in this dataset, we looked for the presence of genetic determinants of AMR. Both horizontally acquired antimicrobial resistance genes (ARGs) and vertically inherited point mutations known to confer resistance or reduced susceptibility to various antimicrobials in *E. coli* were identified. ARGs were exclusively identified in human isolates, except for one captive zoo bird isolate in HAC BAPS cluster 8 (SRR13092475). Overall, human isolates were observed to carry more

AMR genetic determinants compared to bird isolates, including a total of 25 ARGs and five point mutations associated with resistance or reduced susceptibility to 10 different antimicrobial drug classes. In contrast, only three point mutations were identified among the bird isolates, with the exception of the aforementioned captive bird isolate (SRR13092475) carrying an additional five ARGs associated with resistance to four antimicrobial drug classes. Point mutations were more frequent than ARGs, but the implications were less clear. Specifically, *uhpT* E350Q and I355T (Fig. 3a), predicted to confer resistance against fosfomycin and quinolone, respectively[29], were identified in all human and bird isolates, with the multidrug-resistance-associated *marR* S3N point mutation being identified in the majority ($n = 157/162$, 97%) of isolates.

There were 18 unique genotype profiles, including three dominant profiles identified in 80% ($n = 66/83$) of the isolates (Fig. 3b). Correlating ARGs with the phylogeny revealed that the majority (14/16) of isolates within the HAC BAPS 4 had the ARGs *blaDHA-1*, *blaTEM-1*, *dfrA17*, *mph(A)*, *qnrB4*, *sul1* and *tet(A)* (Fig. 1). Among these, *mph(A)*, *sul1*, *blaDHA-1* and *qnrB4* were present on a single contiguous sequence in multiple isolates, the longest of which was 14,961 bp. A BLASTn search of this contiguous sequence revealed 100% coverage and identity with plasmids from multiple *E. coli* strains, *Shigella sonnei* and *S. flexneri* (Supplementary Data 3). Single contiguous sequences containing the four ARGs were identified in 13 isolates, all belonging to BAPS cluster 4. A single point mutation in the quinolone resistance determining region (QRDR) of gyrA, S83L, was present in 60% (43/72) of HACs isolates (though this was not present in BAPS 3) and only 1% (1/ 90) of isolates in BACs (Fig. 1).

The genotypic AMR profile among human isolates was further explored through phenotypic testing. We selected 11 *E. albertii* (HAC $n = 7$, BAC $n = 4$) isolates that captured the lineage and genotypic AMR diversity across the phylogenetic tree and determined their antimicrobial resistance profiles against cefoxitin, ceftriaxone, fosfomycin, tetracycline, ampicillin, ciprofloxacin, chloramphenicol and rifampicin, to review the phenotypic consequences of mutations identified in this study (Figs. 1 and 3). The presence of ARGs *tet(A)*, *blaTEM-1* and *qnrB4* conferred resistance to tetracycline, beta-lactam and fluoroquinolone class antibiotics, respectively (Table 2). The presence of ARG *blaDHA-1* did not confer resistance to the cephalosporin class antibiotics, cefoxitin and ceftriaxone, in this isolate set. Point mutations *uhpA_G97D** and *uhpT_E350Q**, when present together, as well as point mutations in *gyrA_S83L**, were associated with resistance/ decreased susceptibility to their related antimicrobial classes (fosfomycin and fluoroquinolones respectively). Point mutations in *marR_S3N** and *parE_I355T** were present in the majority of isolates tested in this study set, and resistance profiles were consistent across the dataset and impacted by the additional presence of other ARGs or point mutations (Table 2).

### Global contextualisation of *E. albertii* from GB

To place the human and bird *E. albertii* isolates from GB within the global context, we expanded the analysis to include additional isolates retrieved from publicly available data ($n = 475$, "Methods", Supplementary Data 4). A cgMLST tree was generated based on hierarchical clustering of 2513 gene loci. These additional isolates were derived from diverse sources (22% human; 47% Avian ['poultry', 'non-poultry' and 'not defined']; 7% mammal (e.g., livestock, wild species and companion species); 1% food; 2% water and 21% undescribed sources) and locations (18% Americas, 16% Europe, 41% Asia, 1% Africa, 1% Oceania, and remaining 23% unknown). We correlated the position of BAPS clustered isolates from the current study in this broader context (meaning, notably, that BAPS notation is specific to the BACs and HACs groupings of *E. albertii* isolates from GB). We observed that isolates from GB were dispersed across most parts of the cgMLST tree, indicating that these isolates capture much of the known diversity of

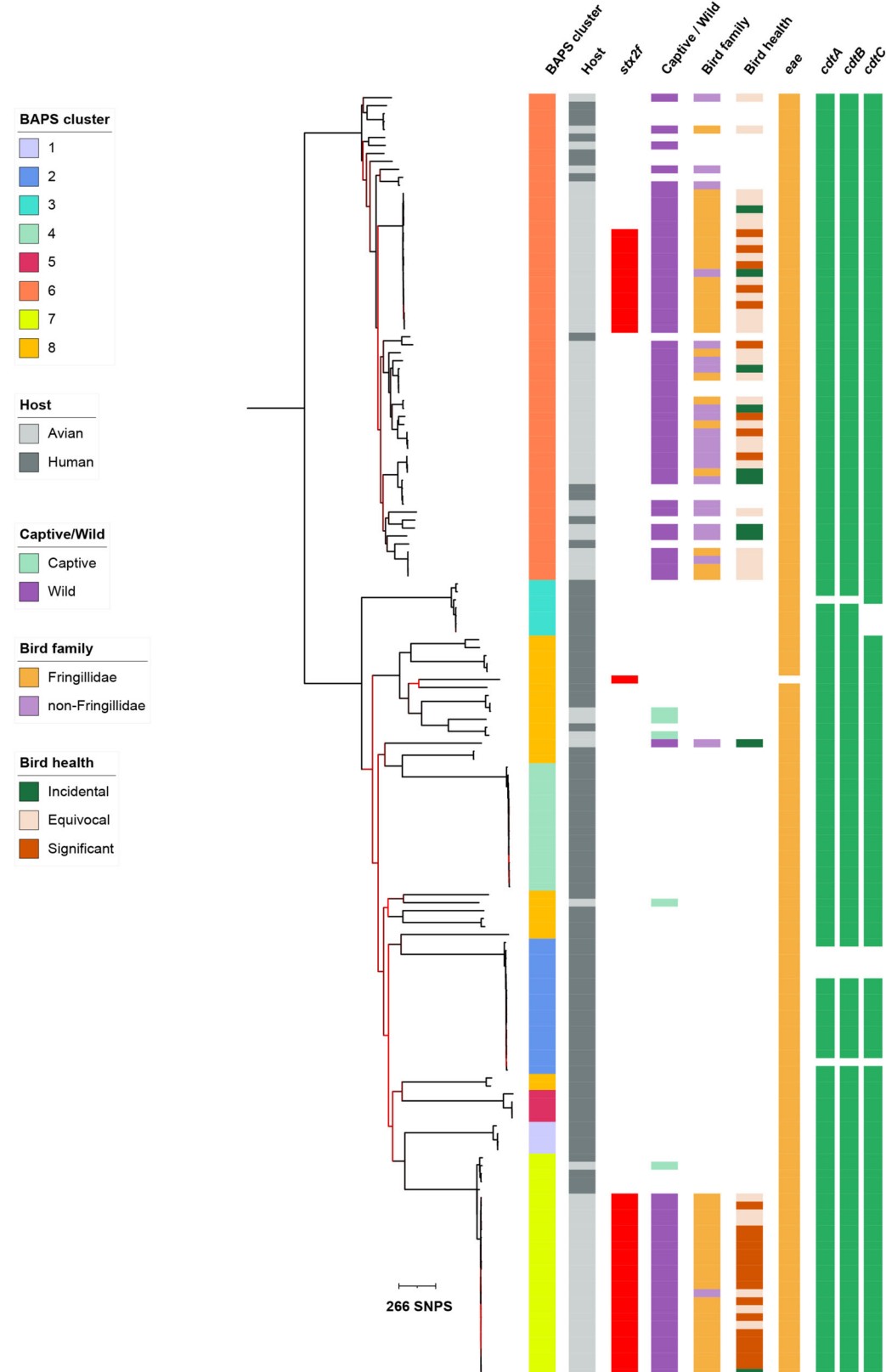

**Fig. 2 | Midpoint rooted maximum likelihood phylogenetic tree of *Escherichia albertii* isolates from Great Britain showing bird host characteristics and virulence-associated genes.** The scale bar is shown in single nucleotide polymorphisms (SNPs). Isolate metadata are displayed on the adjacent tracks according to the inlaid keys, with the presence of virulence-associated genes indicated by a colour block in subsequent tracks (*eae* in yellow, *cdt* genes in green). Phylogenetic branches highlighted in red indicate nodes with low bootstrap support (between 50 and 70%). BAPS = Bayesian Analysis of Population Structure.

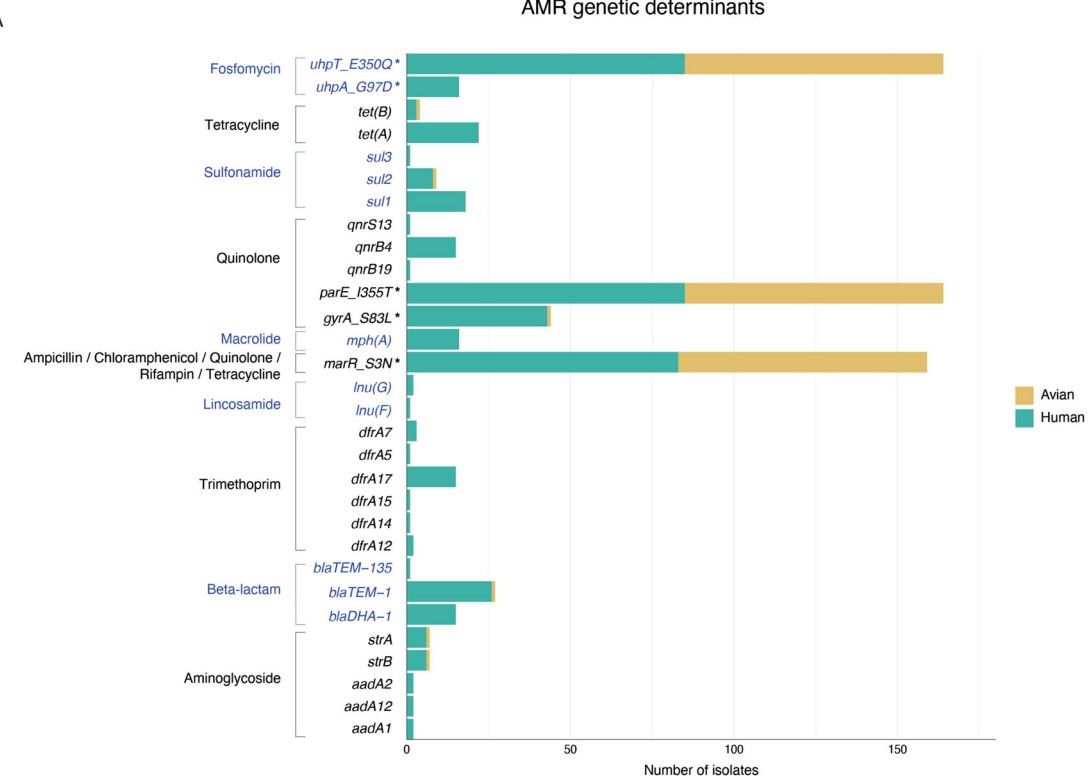

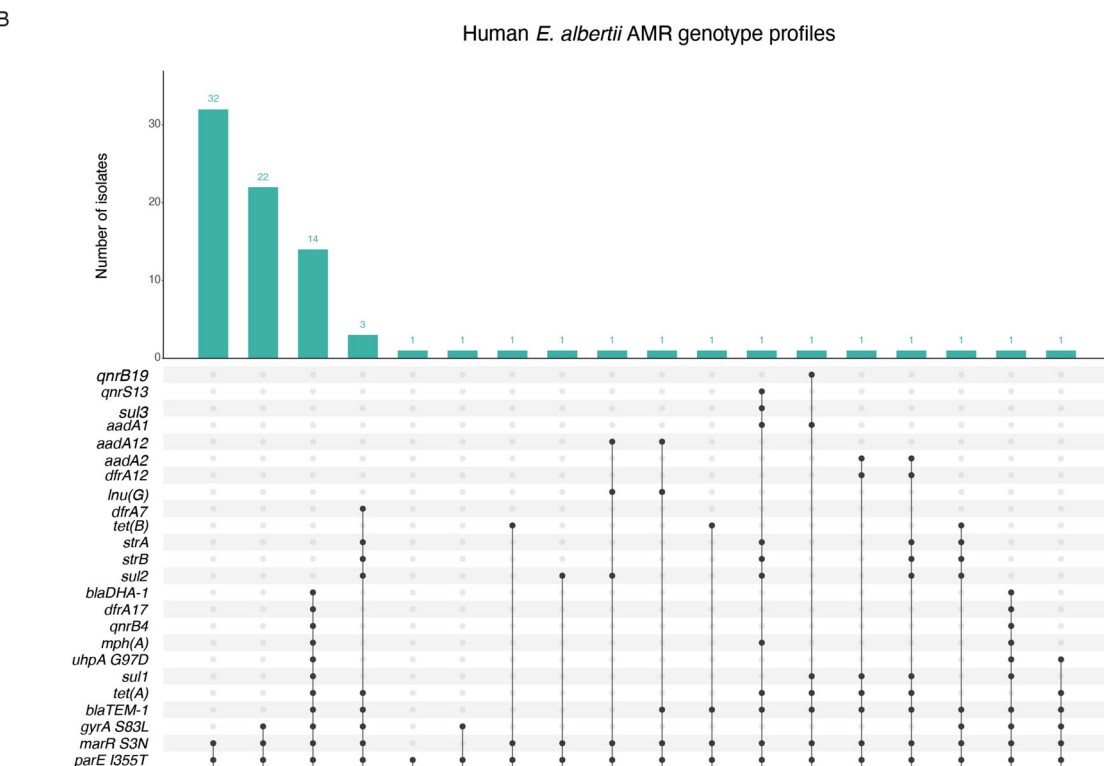

**Fig. 3 | Occurrence of antimicrobial resistance (AMR) among *Escherichia albertii* isolates from Great Britain. A** Stacked barplot demonstrates the number of isolates from birds and humans carrying known AMR genetic determinants. Genetic determinants highlighted with asterisks represent point mutations and different antimicrobial drug classes shown in alternating coloured text. **B** UpSet plot illustrates the prevalence of AMR genotypic profile among human isolates. The combination matrix in the centre panel shows the various genotypic AMR profiles, in which each column represents a unique profile, and each black dot represents the presence of a genetic determinant conferring resistance/reduced susceptibility to a drug class (displayed on the left). The vertical barplot above the matrix shows the number of isolates with a particular genotype, and the number above each bar shows the exact number of isolates with the genotype.

**Table 2 | Antimicrobial resistance phenotypes of 11 *Escherichia albertii* isolates with varied antimicrobial resistance genotypes**

| Sequence Read Archive Accession | Genotype[a] | Minimum inhibitory concentration (µg/mL)[c] | | | | | | | |
| --- | --- | --- | --- | --- | --- | --- | --- | --- | --- |
| | | Cephalosporin | | Fosfomycin | Tetracycline | β-lactam | Fluoroquinolone | Chloramphenicol | Rifampicin |
| | | Cefoxitin | Ceftriaxone | Fosfomycin | Tetracycline | Ampicillin | Ciprofloxacin | Chloramphenicol | Rifampicin |
| SRR12769799 | uhpT_E350Q*, marR_S3N*, parE_I355T* | 8 | 0.047 | 1 | 1.5 | 3 | 0.016 | 6 | 8 |
| SRR12769953 | uhpT_E350Q*, marR_S3N*, parE_I355T* | 6 | 0.047 | 6 | 0.5 | 4 | 0.008 | 3 | 2 |
| SRR13049225 | uhpT_E350Q*, marR_S3N*, parE_I355T* | 6 | 0.047 | 2 | 1 | 4 | 0.006 | 4 | 4 |
| SRR13049237 | uhpT_E350Q*, parE_I355T* | 6 | 0.047 | 1 | 1 | 4 | 0.012 | 6 | 3 |
| SRR11442290 | uhpT_E350Q*, marR_S3N*, parE_I355T*, gyrA_S83L* | 4 | 0.047 | 1.5 | 0.75 | 6 | 0.125 | 3 | 12 |
| SRR15338008 | uhpT_E350Q*, marR_S3N*, gyrA_S83L*, parE_I355T* | 1.5 | <0.016 | 1.5 | 0.38 | 3 | 0.032 | 2 | 4 |
| SRR8981835 | uhpT_E350Q*, marR_S3N*, gyrA_S83L*, parE_I355T* | 1.5 | <0.016 | 1.5 | 0.25 | 3 | 0.032 | 2 | 4 |
| SRR15338057 | blaDHA-1, uhpA_G97D*, uhpT_E350Q*, marR_S3N*, blaTEM-1, gyrA_S83L*, parE_I355T*, qnrB4 | 6 | 0.064 | 12 | 48 | >256 | 0.38 | 1.5 | 4 |
| SRR9050433 | blaDHA-1, uhpA_G97D*, uhpT_E350Q*, tet(A), marR_S3N*, blaTEM-1, gyrA_S83L*, parE_I355T*, qnrB4 | 1 | 0.047 | 8 | 48 | 96 | 0.5 | 3 | 16 |
| SRR11425059 | uhpT_E350Q*, tet(A), marR_S3N*, blaTEM-1, gyrA_S83L*, parE_I355T* | 12 | 0.064 | 2 | 48 | >256 | 0.19 | 4 | 8 |
| SRR3574322 | uhpT_E350Q*, tet(A), marR_S3N*, blaTEM-1, gyrA_S83L*, parE_I355T* | 3 | <0.016 | 1 | 32 | 96 | 0.023 | 4 | 4 |
| Genotype associated with resistance[b] | | blaDHA-1 | | uhpA_G97D*, uhpT_E350Q* | tet(A), tet(B), marR_S3N* | blaTEM-1, blaTEM-135, marR_S3N* | gyrA_S83L*, parC_S57T*, parE_I355T*, qnrB19, qnrB4, qnrS13, marR_S3N* | marR_S3N* | marR_S3N* |

[a]Genes and point mutations (*) found present in isolates in this study.

[b]Genes and point mutations (*) associated with resistance to antimicrobial classes tested in this study.

[c]Underlined MIC determination results highlight MIC breakpoints ((g mL⁻) classed as resistant according to EUCAST guidelines. (https://www.eucast.org/fileadmin/src/media/PDFs/EUCAST_files/Breakpoint_tables/v_11.0_Breakpoint_Table).

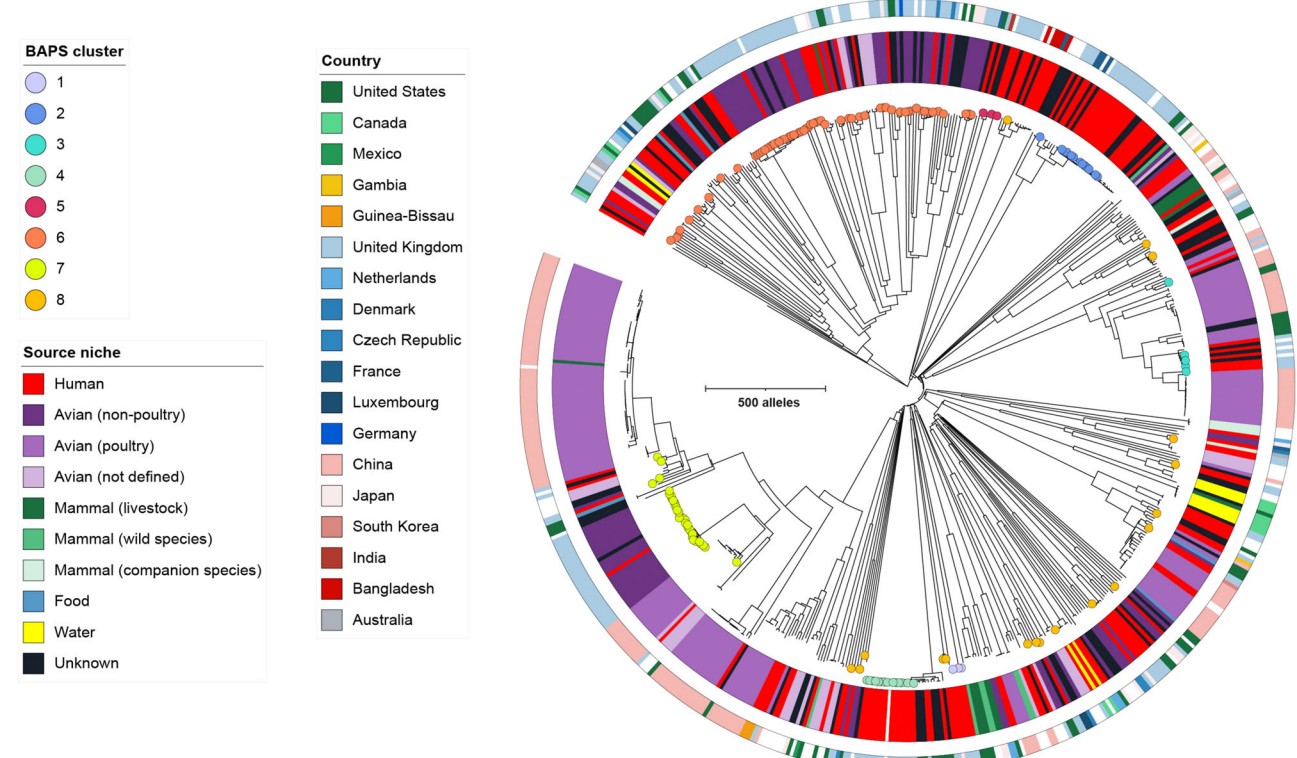

**Fig. 4 | Phylogenetic tree comprising 162 *Escherichia albertii* isolates from the current study and an additional 475 isolates retrieved from EnteroBase.** The tree was constructed based on core genome MLST profiles. Circles at tree tips highlight *E. albertii* isolates from Great Britain under investigation in this study, and the colour of the circles represent the BAPS clusters identified earlier in the study. The thicker inner ring demonstrates the source niche of the isolates, and the thinner outer ring demonstrates the isolate country of origin, all of which are labelled according to the inlaid keys displayed on the left.

*E. albertii.* The cgMLST tree also revealed that while isolates belonging to the HACs BAPS 2, 4 and 5 remained largely within individual clades of the tree alongside other human-derived isolates (Fig. 4), isolates from HAC BAPS 3 clustered with poultry-derived isolates from Asia and the USA. The majority of isolates from the wild BAC BAPS 7 were similarly embedded within a cluster, this time dominated by poultry isolates from Asia. However, isolates from BAC BAPS 6 and HAC BAPS 8 appeared in multiple clades intermixed with isolates derived from various sources. This is consistent with their greater phylogenetic distance relative to other BAPS clusters (particularly the polyphyletic BAPS 8, Table 1) and suggests that the association of these two BAPS clusters as bird- and human-associated may be less clear.

## Discussion

The notification of cases of GI disease caused by *E. albertii* in GB in both humans and animals is currently low compared to other well-established pathogens, such as *Campylobacter* and *Salmonella* species[30,31]. However, it is likely that the number of *E. albertii* diagnoses will increase in line with improvements in molecular diagnostics and the wider adoption of PCR and WGS as tools for GI pathogen surveillance. Thus, analysing current data to understand the potential public health burden, clinical significance, and risk factors in human and animal hosts will guide future research and surveillance.

Although epidemiological follow-up is not conducted for *E. albertii*, the patterns we observed for *E. albertii* infection in people are consistent with similar transmission routes and risk factors as other GI pathogens. Specifically, a similar proportion of reported travel association (31%, 24/83) with other travel-associated *Enterobacteriaceae*, including *Shigella* (19–50% for the years 2005–2014)[32] and *Salmonella* (19–32% for the years 2005–2014)[33]. We also explored whether, like *Salmonella* and *Campylobacter* species[34], zoonotic infection might contribute to disease transmission. Our observations that GB human and bird isolates belonged primarily to host-associated monophyletic groups and had distinct and convergent accessory genome features (e.g., with HACs containing or acquiring ARGs and the occurrence of *stxf2* in BACs) do not support substantial cross-species transmission (i.e., zoonotic or anthroponotic) between birds and humans. The acquisition of ARGs in HACs of *E. albertii*, however, may have been confounded by geography as many patients had recently travelled to Asia, a known risk factor for enteric pathogen ARG acquisition, and where convergent evolution of QRDR mutations is reported[35,36].

Although our data are not supportive of extensive zoonosis for *E. albertii*, the existing evidence supports reinforcing public health messaging. Specifically, the human isolates grouped in the BAC BAPS 6 (*n* = 13) were not very closely related to bird isolates in BAPS 6 (Fig. 1). Comparatively, the human isolates grouped in BAPS 7 (*n* = 4) had higher similarity with avian isolates in BAPS 7, possibly indicating the occurrence of zoonotic transmission (Fig. 1, Table 1). Supplementary feeding of garden birds is a common pastime in GB that results in a close human–wildlife interface[37], and zoonotic infection has been suggested for other bacterial pathogens of wild birds[38,39]. Furthermore, humans infected with isolates belonging to BAC were typically very young or older people, consistent with a bias towards infant infection, previously described for wildlife-associated *Salmonella* Typhimurium and *S.* Enteritidis biotypes[39,40]. Hence, some (*n* = 4) human isolates conceivably represent zoonotic infections, reinforcing the need for good hygiene measures (e.g., hand washing after handling bird feeders) when feeding garden birds[39].

Four of five captive zoo bird isolates clustered within monophyletic subclades of the HAC BAPS8. However, similar to the human/ bird mixing observed within BAPS 6 (see above), the large genomic

divergence among isolates in BAPS 8 is not indicative of direct anthroponotic transmission (Table 1), and there are other potential sources of *E. albertii* infection for captive birds (e.g., diet, wild birds).

Our study did not strongly support evidence of zoonotic infection in contrast to recent studies from China, Japan and the USA that highlighted the potential for foodborne transmission of *E. albertii* to humans via the consumption of poultry[8,24,26]. Incorporating public data revealed that one HAC (BAPS 3) admixed with poultry-associated isolates from China and the USA (Fig. 4), indicating the possibility that *E. albertii* infections may be a foodborne illness linked to eating poultry either domestically or overseas. The cluster supporting potential zoonotic infection from our study (BAPS 7) also encompassed a broader group of poultry isolates, possibly indicating longer-term transmission among wild birds, poultry and humans for some lineages. In contrast, HACs BAPS 2 and BAPS 4 were on long branches without close associations with other hosts or regions (Fig. 3), potentially indicating an unobserved reservoir of infection, either overseas and/or in non-human hosts. The emerging picture of *E. albertii* as a travel-associated pathogen with a potential reservoir in poultry parallels other enteric pathogens, including *Salmonella* and *Campylobacter*[30,31]. Therefore, genomic surveillance of *E. albertii* in more locations and potential reservoir hosts is needed to further elucidate this pathogen's ecology.

Results from this study, combined with the published literature[2,21,24,41,42], indicate that avian hosts are likely to play a larger role in the epidemiology of *E. albertii* than other (e.g., mammalian) hosts. Analysis of publicly available isolates revealed that comparatively few isolates were derived from non-human mammals relative to birds (7% vs 47% respectively). Although public data are not a reflection of representative surveillance, unpublished data from the ZSL provide a similar picture. While the same microbiological protocol has been used across clinical and routine health check samples from a diverse taxonomic range of birds and mammals held in the ZSL zoological collections since 1991, *E. albertii* has only been identified from five captive birds and not from mammals. Furthermore, there have been no confirmed *E. albertii* detections from livestock or wildlife species in disease surveillance conducted by the Animal Plant & Health Agency (APHA) in England and Wales for 23 years. Although there are limitations to the APHA and ZSL *E. albertii* surveillance (e.g., APHA routine microbiology relies primarily on phenotypic and biochemical characterisation meaning *E. albertii* may be present but not detected; the ZSL captive collections are limited to two sites; the ZSL national wild bird surveillance was skewed towards passerines), an absence of isolation from non-human mammals supports a primary avian reservoir. However, targeted surveillance with broad spatial and taxonomic coverage is required to further explore this hypothesis.

Our study also identified implications of *E. albertii* for bird health, with infection being more frequently associated with significant disease in finch than in non-finch species. This is consistent with historical investigations of multiple mortality incidents of finches in Scotland and the USA[2,19] and supports the hypothesis that it acts as a primary pathogen in these birds. This familial bias may relate to host or environmental factors (e.g., differential exposure or susceptibility) as well as pathogen factors. Our data support the latter, with a possible role for differences in virulence determinant components among circulating *E. albertii* strains affecting infection outcomes. Specifically, isolates containing the *stx2f* virulence factor were associated with finch hosts (Fringillidae), and infection in these birds was significantly more likely to be associated with disease. This relationship could not be disentangled further owing to the low occurrence of *stx2f*-bearing strains from non-finch species, but it is possible that finches may act as a reservoir of *stx2f*-positive *E. albertii*, as is hypothesised for garden bird-associated biotypes of *Salmonella* Typhimurium[43]. Infection with *E. albertii* was also inferred as having possible health impacts on other bird species; further surveillance is required to explore this further.

In conclusion, poor molecular diagnostic capabilities for *E. albertii* in human and animal health laboratories mean the true burden of *E. albertii* infection is likely underestimated, and the lack of systematic surveillance data means that clinical severity and exposure risks are largely unknown. However, we leveraged available data to highlight the likely relevance of travel to regions with a high risk of GI infections, including an association with AMR, and a potential zoonotic component that is likely bird-associated, apparently more so with poultry than with wild bird species. To improve surveillance for *E. albertii*, we recommend increased deployment of molecular diagnostics in medical and veterinary laboratories in conjunction with the systematic collection of epidemiological data. Maintaining close collaborations between public health and veterinary institutions is essential to better understand the source, transmission and risks to animal and public health of this recently identified pathogen.

## Methods

### Human isolates and epidemiological data collection

Diagnostic algorithms for the detection of *E. albertii* are not included in the UK Standard Microbiology Investigation of Gastroenteritis protocols used by local hospital diagnostic laboratories (https://www.gov.uk/government/publications/smi-s-7-gastroenteritis-and-diarrhoea). Between 2014 and 2021, isolates from routine gastrointestinal surveillance, including faecal specimens from hospitalised cases or cases in the community, were either submitted to the GBRU at UKHSA from local hospital diagnostic laboratories in England having been misidentified as *Shigella* species or DEC or were cultured from faecal specimens sent to GBRU for molecular testing. At GBRU, bacteria cultured from faecal specimens on MacConkey agar following aerobic incubation overnight were tested for virulence genes that define the different pathotypes of DEC using PCR, including *eae* which is a characteristic of EPEC, STEC and *E. albertii*[7].

All *eae*-positive isolates were genome sequenced, and bacterial identification was confirmed from the genome using a kmer-based approach, as described previously[44]. In total, all 83 isolates identified as *E. albertii* using this approach were included in this study (Supplementary Data 1). Where available, human isolates were linked to demographic data, including age category, gender, and travel history (Supplementary Data 1).

### Bird isolates and epidemiological data collection

Wild bird-derived *E. albertii* isolates ($n = 74$) were obtained through scanning surveillance of dead wild birds conducted by ZSL over the period 2000–2019 inclusive (Supplementary Data 2). Members of the public reported observations of wild bird mortality, typically in the vicinity of garden bird-feeding stations; consequently, the species coverage was predominantly small passerines (e.g., Fringillidae, Paridae, Passeridae, Turdidae) and columbids, which commonly visit peri-domestic habitats in Great Britain. Carcasses were submitted from a subset of mortality incidents for post-mortem examination. Coverage was across Great Britain, although the majority of wild bird submissions and those from which *E. albertii* was isolated were from England (England $n = 63$ isolates, Wales $n = 6$, Scotland $n = 5$). Post-mortem investigations were conducted following a standardised protocol, supported by parasitological and microbiological examination as routine, combined with histological examination and other ancillary diagnostic testing as indicated based on macroscopic abnormalities. Liver and small intestinal tract contents were routinely sampled for microbiological examination using a standardised protocol[43]. Semi-translucent, butyrous, non-lactose fermenting and oxidase-negative colonies of Gram-negative rods to coccobacilli were subjected to an Analytical Profile Index 20E biochemical test (bioMerieux): isolates tentatively identified as *E. albertii* were cryo-archived at −80 °C. Where *E. albertii* was isolated from multiple wild birds examined from the same mortality incident, a single isolate was submitted to GBRU with

two exceptions where two isolates were typed. An available archive of similarly identified *E. albertii* isolates from clinical examinations ($n = 2$) and post-mortem examinations ($n = 3$) of captive birds in the zoological collection at ZSL was also included (Supplementary Data 2). Additionally, a single *E. albertii* isolate was identified from a sample of small intestinal tract contents collected from a dead wild bird examined post-mortem using the UKHSA diagnostic algorithm for human faecal samples.

The inferred significance of *E. albertii* infection to wild and captive zoo bird health (i.e., its likely contribution to the cause of death) was classified as significant, equivocal, or incidental based on the review of the incident history and the pathological, microbiological and parasitological findings for those examined post-mortem (see Supplementary Methods for full definitions). For the two captive zoo birds with *E. albertii* isolated from clinical samples, the history and ancillary diagnostic test results were also appraised to infer likely isolate significance to host health.

## Genome sequencing and quality control
Isolates of *E. albertii* from UKHSA and ZSL were sequenced at GBRU according to previously described protocols[44] and deposited in the Sequence Read Archive (SRA) under the bioproject accession PRJNA315192 with the SRA accession numbers of individual isolates listed in Supplementary Data 1. Short-read sequences were retrieved from the SRA and processed using Trimmomatic v0.38[45] to trim adaptors and filter low-quality bases. FastQC v0.11.6 (https://www.bioinformatics.babraham.ac.uk/projects/fastqc/) and MultiQC v1.7[46] were used to assess the quality of reads.

## Phylogenetic and clustering analysis
Processed reads were mapped to the *E. albertii* strain 1551-2 reference genome (GenBank accession CP025317)[47] using BWA mem v0.7.17[48]. Alignment files were sorted and filtered using the SAMtools suite v1.9–47[49], and PCR duplicates were marked using Picard v2.21.1-SNAPSHOT MarkDuplicates (http://broadinstitute.github.io/picard/). The BCFtools suite v1.9–80[49] was used to identify sequence variants and filter variant files, in which low-quality single nucleotide polymorphisms (SNPs) were removed if mapping quality <60, Phred-scaled quality score <30, read depth <10 and variant allele frequency <0.7.

BCFtools consensus was used to generate reference-based pseudogenomes for each isolate from the filtered SNP variants. Regions containing insertion sequences and phages (identified using the PHASTER web server https://phaster.ca/) were identified from the reference genome and masked using BEDTools v2.28.0 maskfasta[50]. Regions with a read depth of <10 were also masked. The masked pseudogenomes were concatenated and provided as an alignment for Gubbins v2.3.4[51] to identify and mask regions of putative recombination (Supplementary Fig. 2). Following Gubbins, SNP-sites v2.4.1[52] was used to extract variant sites, producing a final SNP alignment of 26,594 bp in length. This SNP alignment was used to construct a maximum-likelihood phylogenetic tree using IQ-TREE v2.0-rc2[53], constructed based on the FreeRate nucleotide substitution, invariable site, and ascertainment bias correction model with 1000 bootstrap replicates. The phylogenetic tree was midpoint rooted and visualised using interactive Tree of Life (iTOL) v6.5[54].

RhierBAPS v1.1.3[55] was used to identify clusters of genetically similar isolates among the SNP alignment, termed Bayesian Analysis of Population Structure (BAPS) clusters.

## Construction of cgMLST tree with publicly available data
To deepen the insights gained from the UKHSA and ZSL *E. albertii* isolates, we analysed their genome sequences in the context of publicly available *E. albertii* sequence data. Specifically, additional publicly available *E. albertii* genome sequences accessible through Enterobase on 7 February 2022 ($n = 475$) were constructed alongside

the data above into a core genome Multi Locus Sequence Type (cgMLST) tree using hierarchical clustering (HeirCC)[56]. Minimal metadata on source and country of origin was extracted from Enterobase alongside HeirCC level classifications and visualised over the unrooted cgMLST tree using interactive Tree of Life (iTOL) v6.5[54]. Metadata on isolate origin was manually curated into the following categories: human, avian (poultry, non-poultry and not defined); mammal (livestock, wildlife and companion species); food, water and undescribed sources.

## AMR and virulence gene analysis
Draft genomes were assembled de novo from processed short-read sequences using Unicycler v0.4.7[57] with –min_fasta_length set to 200. Qualities of the draft assemblies were assessed with QUAST v5.0.2[58] and were all within the assembly quality standards of EnteroBase for *Escherichia*[56]. Prokka v1.13.3[59] was used to annotate draft genome sequences.

The presence of known genetic determinants of AMR was detected using AMRFinderPlus v3.9.3[60] and screened against the Pathogen Detection Reference Gene Catalog (https://www.ncbi.nlm.nih.gov/pathogens/). AMRFinderPlus was run with the organism-specific option for *Escherichia* and screening for both point mutations and genes (with 80% coverage and 90% identity threshold applied). AMR resistance profiles were visualised with UpSetR v2.1.3[61].

The association of known AMR genes with related plasmid sequence were identified by extracting AMR-gene containing contiguous sequences from draft genome assemblies and comparison against the NCBI nonredundant database using MegaBlast.

Detection of virulence genes was performed using ABRicate (https://github.com/tseemann/abricate), by which draft genomes were screened against the Virulence Factor Database with a minimum nucleotide identity of 80% and minimum coverage of 60%. This screen comprised virulence genes associated with *E. albertii* including *stx*, *eae* and *cdtABC* genes that encode Shiga toxin, intimin and CDT.

## Statistical testing
Statistical support for phylogenetic clustering of bird and human isolates was evaluated with chi-square testing on: (1) the proportion of human isolates in individual clusters (Table 1) and (2) associations of human-associated clusters (HACs) and bird-associated clusters (BACs) with patient age (categorised into infant [<2 years], children [2–15 years], adult [16–60 years] and older people [>60 years]). In the finch (Fringillidae) hosts, any significance between the presence of *stx*2f and clinically significant *E. albertii* infection was also evaluated using Fisher's exact test. Adjusted and strata-specific odds ratios for the effect of bird family on the association between *stx2f* presence and inferred significant disease were conducted using the Mantel–Haenszel Test. All statistical tests were performed using R v4.0.3.

## Phenotypic antimicrobial resistance testing
Minimum inhibitory concentration (MIC) determination was carried out using Lioflichem→ MIC test strips (Lioflichem, Italy) following the manufacturer's guidelines. Bacterial inoculum for MIC testing was prepared, following the EUCAST guidelines for *Enterobacterales* standard broth microdilution (https://www.eucast.org/fileadmin/src/media/PDFs/EUCAST_files/Breakpoint_tables/v_11.0_Breakpoint_Tables.pdf) and was spread on Mueller–Hinton Agar plates (Bio-Rad, France) using sterile cotton swabs after which the MIC test strip was applied. Plates were incubated at 37 °C for 18 h before the readings were recorded.

## Reporting summary
Further information on research design is available in the Nature Portfolio Reporting Summary linked to this article.

## Data availability

Individual accession numbers for isolates used in this study are available in Supplementary Data 1, 2, and 4. Phylogenetic trees from this study have been deposited in figshare (https://doi.org/10.6084/m9.figshare.20894854.v1). The authors recognise that this study opens up important further avenues for functional research of *E. albertii* and are happy to make isolates available on request.

## Code availability

No custom code was used in the analysis of this data.

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

## Acknowledgements

We thank the members of the public and participants in the British Trust for Ornithology's Garden BirdWatch, who assisted with reporting observations of wild bird mortality to ZSL as part of a national wild bird disease surveillance programme, initially the Garden Bird Health *initiative* (2005–2012) followed by the Garden Wildlife Health project (2013–2019) (www.gardenwildlifehealth.org). We thank the veterinarians who conducted some of the wild bird post-mortem examinations included in this study (including Katie Beckmann, Lydia Franklinos, Joseph Heaver and Vicky Wilkinson) and Ed Fullick from the Animal & Plant Health Agency for sharing information. Financial support for the Garden Bird Health initiative came from the Birdcare Standards Association, British Trust for Ornithology, British Veterinary Association Animal Welfare Foundation, CJ Wildbird Foods, Cranswick Pet Products, UK Department for the Environment Food & Rural Affairs (Defra) and Welsh Government through the Animal & Plant Health Agency's (APHA) Diseases of Wildlife Scheme (DoWS) Scanning Surveillance Programme (Project ED1058), Gardman Ltd, Institute of Zoology, Royal Society for the Protection of Birds and the Universities Federation for Animal Welfare. Financial support for the Garden Wildlife Health Project came in part from the Department for Environment, Food and Rural Affairs (Defra), the Welsh Government and the Animal and Plant Health Agency (APHA) Diseases of Wildlife Scheme (DoWS); and from the Banister Charitable Trust, Esmée Fairbairn Foundation, Garfield Weston Foundation and the Universities Federation for Animal Welfare. IoZ staff receive financial support from Research England. This work was supported by a Medical Research Council Grant (MR/R020787/1, K.S.B.) and Biotechnology and Biological Resources Council Grant (BB/V009184/1, K.S.B.). K.S.B, D.R.G. and C.J. are affiliated with the National Institute for Health and Care Research Health Protection Research Unit (NIHR HPRU) in Gastrointestinal Infections at the University of Liverpool in partnership with the United Kingdom Health Securities Agency (UKHSA), and the University of Warwick. The views expressed are those of the author(s) and not necessarily those of the NHS, the NIHR, the Department of Health and Social Care, or the UKHSA.

## Author contributions

According to the Contributor roles taxonomy, author contributions were as follows: Conceptualisation—K.S.B., B.L. and C.J.; Data curation—B.L., S.K.J., S.K.M., S.S., D.G., C.C. and P.M.D.S.; Formal analysis—K.S.B., R.J.B., C.C. and P.M.D.S.; Funding acquisition—K.S.B., A.A.C., B.L. and C.J.; Investigation—K.S.B., A.A.C., B.L., K.S.-M., S.S., C.J., R.J.B., S.K.J., S.K.M., P.M.D.S. and C.C.; Methodology—K.S.B., A.A.C., B.L., K.S.-M., S.K.J., S.K.M., S.S., D.G. and R.J.B.; Project administration—K.S.B., C.J. and B.L.; Resources—A.A.C., B.L., C.J. and D.G.; Software—R.J.B., Supervision—K.S.B. and C.J.; Validation—D.G. and R.J.B.; Visualisation—R.J.B.; Writing original draft—C.J., R.J.B., B.L. and K.S.B. Writing review and editing—All.

## Competing interests

The authors declare no competing interests.

## Ethics statement

Samples were collected during post-mortem examination of wild birds found dead or euthanased for welfare reasons under the Veterinary Surgeons Act 1966. Samples from humans. For data relating to isolates from the United Kingdom Health Security Agency: No individual patient consent was required or sought as UKHSA has the authority to handle patient data for public health monitoring and infection control under section 251 of the UK National Health Service Act of 2006 (previously section 60 of the Health and Social Care Act of 2001.
