## [Peer Review File · Nature Communications]

The genomic epidemiology of *Escherichia albertii* infecting humans and birds in Great BritainREVIEWER COMMENTS

Reviewer #1 (Remarks to the Author):

This manuscript presents an NGS based study on *Escherichia albertii*. *E. albertii* is often mis-diagnosed and is therefore under reported. Due to it harbouring AMR genes (and SNPs) and many virulence factors it can be of public health importance. The manuscript is well written, collection is somewhat limited, but seems to be large enough to draw reasonable conclusions, methods and analyses are sound and the conclusions are supported by the data.

Major comments

The overlap of the human and bird data could be better to properly assess the role of zoonotic and anthroponotic infections. Now the human isolates are from 2014 to 2021 and the bird isolates from 2000 to 2019 therefore some of the potential transmission from birds to humans and vice versa might be missed. Given the scarcity of the data, the roles of the human and birds on each other's infections cannot be assessed reliably. The authors should more clearly discuss the limitations of the study.

Minor issues

Abstract

Lines 20-21. The authors could add what *E. albertii* is mostly misidentified as.

Lines 38-40. The authors could add why *E. albertii* is mis-identified. What are the phenotypic tests identifying? What is the marker? As far as I know the diarrhoeagenic *E. coli* are not a monophyletic group.

43-46. It should be clarified what surveillance this is; random collection of all "*E. coli*", all blood isolates or something else.

Line 82 Is something missing after e.g. at the end of the line? At least there is an extra parenthesis.

Line 96. Is the comparison of AMR profiles with phenotypic or genetic data?

Lines 191-192 Why use Unicycler if short reads. Why not Shovill/Spades?

Line 208 The identity cut-off together with minimum coverage of 60% will result in many not closely related hits. Could the authors explain why these cut-offs were used?

Line 215 What are HAC and BAC? I think it's explained later, but this is the first mention.

Lines 249-254. The evaluation of the *E. albertii*

Lines 280-281. BAPS cluster 8 seems to be the bin cluster where BAPS but genomes it can't really assign. That's why it's also polyphyletic. The authors shouldn't concentrate too much on the (statistical) analyses done with it.

Lines 288-294. The age groups. Why these age groups were used. The authors should give some rationale on the selection. Since there is association to e.g. travel would a more detailed division give more informative results? Same goes for the age groups associated with BACs.

Line 325. "The genotypic AMR profile among human isolates was further explored through phenotypic testing."

This paragraph does not really tell anything about phenotypic resistance and how the genotypic results correlate with phenotypic ones. This is written in the next paragraph.

Line 348 Add line break?

Lines 362-363 Still it seems that BAPS cluster 8 is the bin cluster. Therefore, it's to be expected that it's all over the tree. On the other hand, BAPS cluster 6 looks to be fine and still monophyletic in the global tree.

Lines 396-398 The authors should discuss why the isolates are not phylogenetically closely related. In my opinion the study design i.e. the strain collection may be the major reason. Larger studies might be needed to confirm the current results.

Lines 412-415. In a polyphyletic cluster, spanning multiple sections of the tree, diversity is expected to be high. This should not be directly compared to monophyletic clusters.

Lines 446-448. The authors conclusion of birds being the most common host seems to be well supported, but are the samples sent to the ZSL and APHA biased in some way? I would guess that a global collection from wild animals could give different results. I may have misunderstood the role and function of ZSL since it's really clarified in the text.

Reviewer #2 (Remarks to the Author):

The manuscript by Bengtsson titled "The genomic epidemiology of *Escherichia albertii*" describes the sequencing and comparative analysis of a number of *E. albertii* isolates from both humans and birds. The authors identify that there are clades that are associated with either birds or humans, suggesting that there may be some host specialization, but this specialization was not absolute with avian and human isolates being phylogenomically similar. There was limited association with age, travel or host. A range of classical *E. coli* virulence factors were identified in *E. albertii*, including the intimin (*eae*) and Shiga toxin (*stx*) genes. Further analysis by wgMLST including all the identified *E. albertii* isolates identified more mixing of isolates from bird, human and isolates from other sources, suggesting some, but again not absolute host specialization. The numbers of isolates in the study in general is relatively low, and even lower when divided into bird and human isolates, which diminishes enthusiasm for the study, as well as technical issues outlined below.

Major Comments:

Many of the analyses are focused on the assembled data (AMR and virulence), however the assembled data this is based on is no public. The authors have released the primary reads, and some analyses are based on these reads but the assembled data must be released. There are many ways the assembly can differ, even with the included methodology and thus these should be released.

It is unclear if the monophyletic designation of the isolates is based entirely on the BAPS analysis? Visual inspection of the inferred phylogeny in figure 1 would suggest that there are additional phylogenetic distinctions in the BAPS group at the top (maybe BAPS cluster 6 - the colors in the Figure were difficult to distinguish in some cases). The authors could include a SNP threshold or some other value to confirm/validate the BAPS analysis.

The use of Gubbins on such a diverse set of isolates likely will result in the over-conservative removal of significant genomic data. The authors should explore if these groupings are valid without the use of Gubbins.

How were the 11 isolates selected for antimicrobial determination (line 337)? This seems like ~ 1 per BAPS cluster, which would not be significant to be able to make any inferences to the remainder of the cluster?

Figure 4 – the inclusion of the collection of all *E. albertii* isolates is excellent; however it is unclear why the authors used cgMLST here and not total genomic comparison as in Figure 1. The number of genomes could easily be analyzed using the described methodology and provide more robust data analysis.

Such clearly defined groups from different hosts would allow a direct genomic comparison of which genes/genomic regions may be responsible for this host speciation. Possibly

The discussion is much too long and can be shortened by at least 25%.

Minor Comments:

Supplementary Figure S1 does not add much to the manuscript and can be removed as it is described in the text already

Table S3 – it is unclear what point the authors are making with this table. There are 6/14 isolates in one of the monophyletic clades that share a 14Kb genomic region that confers antimicrobial resistance. Maybe a figure of the structure of this 14kb region would help the understanding. This seems incomplete.

Line 44- increase relative to what? Rephrase sentence to indicate that there is the possibility to identify via PCR/WGS that was not present before. The authors are hypothesizing that there were *E. albertii* previously, but this is not known.

The authors identify the intimin gene (*eae*) in the majority of the isolates. Can they elaborate if the rest of the LEE region is present and functional? Additionally, is the Shiga toxin produced? This would provide some additional information beyond the genomics.

BACs is always a poor abbreviation in a genomics paper as it suggests the old days when BACs were the cloning vectors.

Author response (in bold throughout):

We would like to thank the editor and the reviewers for their valuable time and appreciate their helpful, constructive comments on the manuscript. We believe in addressing the suggestions and comments raised, the manuscript has been significantly improved, and we hope the reviewers enjoy reading the revised version.

NL: Refers to New Line numbers in the revised manuscript (clean version).

OL: Refers to Original Line numbers in the original submission.

REVIEWER COMMENTS

Reviewer #1 (Remarks to the Author):

This manuscript presents an NGS based study on *Escherichia albertii*. *E. albertii* is often mis-diagnosed and is therefore under reported. Due to it harbouring AMR genes (and SNPs) and many virulence factors it can be of public health importance. The manuscript is well written, collection is somewhat limited, but seems to be large enough to draw reasonable conclusions, methods and analyses are sound and the conclusions are supported by the data.

Major comments

The overlap of the human and bird data could be better to properly assess the role of zoonotic and anthroponotic infections. Now the human isolates are from 2014 to 2021 and the bird isolates from 2000 to 2019 therefore some of the potential transmission from birds to humans and vice versa might be missed. Given the scarcity of the data, the roles of the human and birds on each other's infections cannot be assessed reliably. The authors should more clearly discuss the limitations of the study.

The reviewer is right that the available archives of this poorly identified pathogen are not as comprehensive as we would want them to be, but we consider the overlap and extent of these collections to be a valuable exploration of the possibility of transmission. We agree with the reviewer that we are not able to comprehensively confirm or refute the possibility of infection with this dataset (and our concluding that more work is needed in the area) and have made this limitation more explicit in the discussion (see OL418-420, NL 446 – 448).

Minor issues

Abstract

Lines 20-21. The authors could add what is *E. albertii* mostly misidentified as.

We have added the reviewer's suggestion in the abstract at NL 21 'as pathotypes of diarrhoeagenic *Escherichia coli* and *Shigella* species'

Lines 38-40. The authors could add why *E. albertii* is mis-identified. What are the phenotypic tests identifying? What is the marker? As far as I know the diarrhoeagenic *E. coli* are not a monophyletic group.

***E. albertii* are mis-identified as *Shigella* species because they have metabolically and biochemically similar laboratory profiles along the diagnostic microbiology pathway, for example the inability to ferment lactose or decarboxylate lysine (see Bergey's manual for full elaboration). We have added this detail to the Introduction (NL 39 – 40). *E. albertii* are mis-identified as enteropathogenic *E. coli* (EPEC) because they have the**

eae gene, which is the diagnostic marker for the EPEC group. This is explained in detail in the Introduction (NL 50 – 53).

43-46. It should be clarified what surveillance this is; random collection of all “E. coli”, all blood isolates or something else.

It is the surveillance of gastrointestinal pathogens isolated from individuals with gastrointestinal symptoms presenting to primary healthcare. This has been clarified in the methods (NL 105).

Line 82 Is something missing after e.g. at the end of the line? At least there is an extra parenthesis.

We thank the reviewer for pointing this out and have amended this.

Line 96. Is the comparison of AMR profiles with phenotypic or genetic data?

This is with comparison of genotypic AMR profiles. This has been updated in NL 97.

Lines 191-192 Why use Unicycler if short reads. Why not Shovill/Spades?

Unicycler is a SPAdes optimiser that performs *de novo* assembly for both long and short reads. Both Shovill and Unicycler use SPAdes as the underlying assembler.

Line 208 The identity cut-off together with minimum coverage of 60% will result in many not closely related hits. Could the authors explain why these cut-offs were used?

The screening of virulence genes was performed with a minimum of 80% sequence identity and 60% coverage, which was the flexibility required to reflect the PCR positivity for this gene among the *E. albertii* isolates. For the *eae* gene, the percentage nucleotide identity with respect to EPEC or EHEC (which is what is included in the Virulence Factor Database) can be as low as 55% (1). When we performed the analysis with a minimum of 90% sequence identity and 85% coverage, the *eae* gene was detected in only three isolates. When rerunning the analysis with 80% sequence identity and 60% coverage *eae* was detected in all but one isolate, with identity ranging between 84 – 92% and coverage between 60-100%. To make this diversity clearer, we have included a new Supplementary Figure to the manuscript which captures additional genetic analyses performed or visualised in response to these reviewer comments (see Supplementary Figure 2) and modification to the text on this point at NL303.

Line 215 What are HAC and BAC? I think it's explained later, but this is the first mention.

This has been amended and HAC and BAC are explained in NL282 – 284.

Lines 249-254. The evaluation of the *E. albertii*

This comment appears to be incomplete?

Lines 280-281. BAPS cluster 8 seems to be the bin cluster where BAPS butts genomes it can't really assign. That's why it's also polyphyletic. The authors shouldn't concentrate too much on the (statistical) analyses done with it.

We agree with the reviewer that BAPS8 is the cluster capturing diverse organisms without apparent correlation to the phylogenetic tree; a common feature among

population genetic clustering software packages. We present the statistical analysis of this alongside the others for completeness, but refrain from making inferences in the text as to its significance for this reason. Where appropriate we have been more explicit about this in the text to address the reviewer's comment – see NLs 372 – 373 and Table 1.

Lines 288-294. The age groups. Why these age groups were used. The authors should give some rationale on the selection. Since there is association to e.g. travel would a more detailed division give more informative results? Same goes for the age groups associated with BACs.

The age groups were used as we were trying to ascertain possibly transmission pathways for this poorly characterised pathogen. The age groups reflect those that were previously effective in identify transmission routes in other public health genomic epidemiology studies of enteric pathogens. Specifically, child-wildlife associated transmission of salmonellosis (see references 57 and 58) and shigellosis among men who have sex with men (see PMID: 25936611).

Line 325. "The genotypic AMR profile among human isolates was further explored through phenotypic testing."

This paragraph does not really tell anything about phenotypic resistance and how the genotypic results correlate with phenotypic ones. This is written in the next paragraph.

We thank the reviewer for pointing out this misplacement of text which has been moved to the head of the appropriate (next) paragraph.

Line 348 Add line break?

We thank the reviewer for spotting this and have added a line break here.

Lines 362-363 Still it seems that BAPS cluster 8 is the bin cluster. Therefore, it's to be expected that it's all over the tree. On the other hand, BAPS cluster 6 looks to be fine and still monophyletic in the global tree.

We agree with the reviewer and feel we have made this clear in the text, Figure 3, and Table 1.

Lines 396-398 The authors should discuss why the isolates are not phylogenetically closely related. In my opinion the study design i.e. the strain collection may be the major reason. Larger studies might be needed to confirm the current results.

Thank you – we agree this is an important point. We have added clarification to the text and highlighted the limitations of the study with respect to sample size (NL 428 – 430).

Lines 412-415. In a polyphyletic cluster, spanning multiple sections of the tree, diversity is expected to be high. This should not be directly compared to monophyletic clusters.

We thank the reviewer for this point, which they have raised several times and has been addressed accordingly as described above.

Lines 446-448. The authors conclusion of birds being the most common host seems to be well supported, but are the samples sent to the ZSL and APHA biased in some way? I would guess that a global collection from wild animals could give different results. I may have misunderstood the role and function of ZSL since it's really clarified in the text.

We thank the reviewer for raising this query. The Institute of Zoology (IoZ) is the conservation science arm of the Zoological Society of London, which also operates ZSL London Zoo and ZSL Whipsnade Zoo. IoZ has several long running wildlife health surveillance programs, including coordination of the Garden Wildlife Health project; a national scanning surveillance program for multiple wildlife species including garden birds. As such, data from the microbiology laboratories at ZSL offer a valuable and rare opportunity for inference regarding *E. albertii* occurrence in both captive and wild animals using consistent isolation protocols over several decades.

The APHA is our national governmental organisation that conducts scanning and targeted surveillance activities for livestock and some wild animal species. While there is likely some bias within the samples tested, this is the agency with the widest national animal health surveillance responsibility in GB, so we believe reference to their unpublished data remains relevant and informative. We have amended the text from APHA to emphasise that this lack of detection may not reflect absence of infection, due to the methods employed.

Thus, while these collections and unpublished data provide the best view of *E. albertii* in birds in Great Britain, it is true there is potential bias as the captive wild bird isolates derive from only two zoological collections, and the free-living wild bird surveillance has taxonomic skew towards small passerines, which is why we highlight the need for extended animal surveillance, in both geographic and taxonomic scope, to further resolve the host range and diversity of this bacterium.

To address the reviewer's concern, we have made the following changes to the text in the results: where we provide additional detail on taxonomic skew and possible bias in wild bird surveillance (see NL 121 - 125), and the Discussion: which make these limitations and their implications more explicit (see NLS 434 - 448).

Reviewer #2 (Remarks to the Author):

The manuscript by Bengtsson titled "The genomic epidemiology of *Escherichia albertii*" describes the sequencing and comparative analysis of a number of *E. albertii* isolates from both humans and birds. The authors identify that there are clades that are associated with either birds or humans, suggesting that there may be some host specialization, but this specialization was not absolute with avian and human isolates being phylogenomically similar. There was limited association with age, travel or host. A range of classical *E. coli* virulence factors were identified in *E. albertii*, including the intimin (*eae*) and Shiga toxin (*stx*) genes. Further analysis by wgMLST including all the identified *E. albertii* isolates identified more mixing of isolates from bird, human and isolates from other sources, suggesting some, but again not absolute host specialization. The numbers of isolates in the study in general is relatively low, and even lower when divided into bird and human isolates, which diminishes enthusiasm for the study, as well as technical issues outlined below.

Major Comments:

Many of the analyses are focused on the assembled data (AMR and virulence), however the assembled data this is based on is no public. The authors have released the primary reads, and some analyses are based on these reads but the assembled data must be released. There are many ways the assembly can differ, even with the included methodology and thus these should be released.

We thank the reviewer for this suggestion and have deposited the assembled data onto the public database, the accession numbers for the assemblies are listed in Supplementary Table 1 and 2.

It is unclear if the monophyletic designation of the isolates is based entirely on the BAPS analysis? Visual inspection of the inferred phylogeny in figure 1 would suggest that there are additional phylogenetic distinctions in the BAPS group at the top (maybe BAPS cluster 6 - the colors in the Figure were difficult to distinguish in some cases). The authors could include a SNP threshold or some other value to confirm/validate the BAPS analysis.

The use of Gubbins on such a diverse set of isolates likely will result in the over-conservative removal of significant genomic data. The authors should explore if these groupings are valid without the use of Gubbins.

The tree was used for monophyly confirmation (as this is the only technique that defines phylogeny). We also calculated average pairwise distances in clusters based on the SNP alignment (Table 1). BAPS clustering was used as a secondary classification tool for the groups, based on sequence alignment alone, and we further present a third classification (cgMLST) analysis of the isolates in a global context. We feel these four different systems (the latter two of which show genomic relationships ignorant of recombination) and our discussion and comparison of them sufficiently describe the population structure and relationships of the isolates in this study.

However, to serve the reviewer and reader's potential interest in recombination among these genomes, we have now included a visualisation of detected recombination in Supplementary Figure 2 and referenced this in the text.

How were the 11 isolates selected for antimicrobial determination (line 337)? This seems like ~ 1 per BAPS cluster, which would not be significant to be able to make any inferences to the remainder of the cluster?

The 11 isolates were selected to cover the breadth of the AMR genotypes detected (as presented in Table 2). This has been clarified in a new title for Table 2.

Figure 4 – the inclusion of the collection of all *E. albertii* isolates is excellent; however it is unclear why the authors used cgMLST here and not total genomic comparison as in Figure 1. The number of genomes could easily be analyzed using the described methodology and provide more robust data analysis. Such clearly defined groups from different hosts would allow a direct genomic comparison of which genes/genomic regions may be responsible for this host speciation.

There are various reasons why this analysis was conducted with cgMLST rather than a SNP-alignment and maximum likelihood inference as in the first figure. Firstly, as the reviewer highlighted, cgMLST is resilient to recombination so this analysis supporting the BAPS grouping highlighted the robustness of the groupings. Secondly, the deep branching among the cgMLST analysis suggests a highly diverse population which would likely result in a substantial computational task for phylogenetic inference without bringing increased resolution to the genomic groupings of the organisms.

The discussion is much too long and can be shortened by at least 25%.

We thank the reviewer for highlighting this issue of readability and have made non substantive edits to the discussion for brevity (see edits throughout). These have resulted in a 15% reduction to the text which we feel addresses this issue.

Minor Comments:

Supplementary Figure S1 does not add much to the manuscript and can be removed as it is described in the text already

We respectfully disagree with the reviewer and elect to leave the Supplementary Figure in place (see below).

Table S3 – it is unclear what point the authors are making with this table. There are 6/14 isolates in one of the monophyletic clades that share a 14Kb genomic region that confers antimicrobial resistance. Maybe a figure of the structure of this 14kb region would help the understanding. This seems incomplete.

We feel there is little value adding a supplementary figure to this supplementary table as the sequences in question are not novel and are publicly available according to the accessions in the table. However, to make the accessibility of this data more explicit, we have edited the headings and also added a specific accession for one of the assemblies from this study (see edits to Table S3).

Line 44- increase relative to what? Rephrase sentence to indicate that there is the possibility to identify via PCR/WGS that was not present before. The authors are hypothesizing that there were *E. albertii* previously, but this is not known.

We thank the reviewer for highlighting this lack of clarity and have changed the text accordingly (NLs 44 – 47). Please note the trend is also shown in Supplementary Figure 1.

The authors identify the intimin gene (*eae*) in the majority of the isolates. Can they elaborate if the rest of the LEE region is present and functional? Additionally, is the Shiga toxin produced? This would provide some additional information beyond the genomics.

We thank the reviewer for highlighting this possible addition to the study and have added the mapping coverage to the LEE region in Supplementary Figure 2 and added this to the methods (NL214 - 215, and further particulars in Supplementary Methods). Elaborating Shiga toxin in the laboratory for the sake of proving it is being produced in animals that are dying from a pathogen containing it (particularly where there is a known association) seems like an unnecessary high-risk experiment that is not required for the conclusions of this study. As such, we have decided not to undertake this experiment in response to the reviewer's comment.

BACs is always a poor abbreviation in a genomics paper as it suggests the old days when BACs were the cloning vectors.

Thank you – we have defined BACs in the context of this study in NL220.

REVIEWER COMMENTS

Reviewer #1 (Remarks to the Author):

The authors have adequately answered the reviewers comments.

Reviewer #2 (Remarks to the Author):

The revised manuscript by Bengtsson et al. with title "The genomic epidemiology of *Escherichia albertii*" describing the sequencing and comparative analysis of a number of *E. albertii* isolates from both humans and birds is much improved with greater clarity and increased required public data release. There remain a number of significant issues that should be addressed.

1) There remain significant issues with the data release as the accession numbers provided in the tables do not correspond to the same isolates in the same rows.

For example: Row 8 in Supplementary Table 1 is

Raw Read Accession : SRR11425059 = *Escherichia albertii* 912272 (which then links to GenBank accession number AATKFN010000000)

Assembly Accession: JAQAMR000000000 = *Escherichia albertii* strain 085 (and this has no link to an SRA submission)

The lack of linkage between these two accession numbers provided seems to suggest that something is wrong between these two columns. Not all rows were checked, but upon spot checking this seemed to be a systemic issue and needs to be addressed as each row I did check had issues.

2) Table S1 does not appear to include unique identifiers for each of the isolates, which would prevent anyone from reading the paper and looking at the data and being able to track back to this primary data.

3) The authors have clarified Figure 1 with colour that there is lack of general support (bootstraps between 50-70), which the authors are suggesting are supported with BAPS and wgMLST. It should be clearly stated that these groupings are not well supported. Additionally in Table 1, barely ½ of the isolates are included in these BAPS or other clusters. While this is a novel finding, the utility of a typing system that can include 50.6% of the isolates has limited utility overall. While in general, I agree with the rebuttal, the data is very diverse, which limits what the authors can say and interpret from it.

4) Regarding the 11 isolates selected for AMR analysis. It is not clear that these are representative of each of the genotypes associated with the presence of various AMR genes. This should be made explicitly clear.

5) The authors have completed an *in silico* LEE analysis and identified that this region is present at a high level of similarity in only a very small number of isolates, and this data is buried in supplemental Figure 2 (and additionally interesting that the LEE and *eae* homology are not congruent). The presence and function of *eae/intimin* is suspect without at least some of the T3SS, yet the authors elect not to mention this in the body of the manuscript. Additionally, in any of the isolates that have a complete LEE, is it functional? It is understood that the authors do not wish to complete functional assays on the Shiga toxin, but the a functional LEE would be a significant in a non-*E. coli* species, especially considering that a functional region is lacking in most isolates.

Author response (in bold throughout)

REVIEWER COMMENTS

The revised manuscript by Bengtsson et al. with title “The genomic epidemiology of *Escherichia albertii*” describing the sequencing and comparative analysis of a number of *E. albertii* isolates from both humans and birds is much improved with greater clarity and increased required public data release. There remain a number of significant issues that should be addressed.

1) There remain significant issues with the data release as the accession numbers provided in the tables do not correspond to the same isolates in the same rows. For example: Row 8 in Supplementary Table 1 is Raw Read Accession : SRR11425059 = *Escherichia albertii* 912272 (which then links to GenBank accession number AATKFN010000000)

Assembly Accession: JAQAMR000000000 = *Escherichia albertii* strain 085 (and this has no link to an SRA submission)

The lack of linkage between these two accession numbers provided seems to suggest that something is wrong between these two columns. Not all rows were checked, but upon spot checking this seemed to be a systemic issue and needs to be addressed as each row I did check had issues.

This was a data linkage issue that has since been corrected.

2) Table S1 does not appear to include unique identifiers for each of the isolates, which would prevent anyone from reading the paper and looking at the data and being able to track back to this primary data.

Unique identifiers for each of the isolates in this study are referred to using SRA accessions which are provided per isolate in Table S1 and readily downloadable from the public Sequence Read Archive. We also deposited the phylogenetic tree files (Newick) published in this study in Figshare, which are labelled with SRAs (SNP tree) and Enterobase IDs (cgMLST), all of which are cross referenced in the Supplementary Tables to facilitate maximum user access to the data.

3) The authors have clarified Figure 1 with colour that there is lack of general support (bootstraps between 50-70), which the authors are suggesting are supported with BAPS and wgMLST. It should be clearly stated that these groupings are not well supported. Additionally in Table 1, barely ½ of the isolates are included in these BAPS or other clusters. While this is a novel finding, the utility of a typing system that can include 50.6% of the isolates has limited utility overall. While in general, I agree with the rebuttal, the data is very diverse, which limits what the authors can say and interpret from it.

BAPS groupings of HACs and BACs isolates were developed in this study to facilitate discussion of the relationships among *E. albertii* collected under a known sampling framework (i.e. isolates from GB), so in Table 1 only these 162 isolates are described. We have changed the title of Table 1 to make this clearer and have also added this detail in NL366.

The function of the BAPS analysis was to provide a parallel exploration of population structure (in addition to the ML phylogeny and cgMLST analysis) and provide a nomenclature to discuss our results. It is not intended to function as a public health surveillance typing system.

While we agree with the reviewer that there is poor support for the phylogenetic grouping of HAC BAPS8, all other clusters have good boot strap for a distinct MRCA relative to other groups. The poor clustering and unclear evolutionary history of BAPS8 is clear in the manuscript and was made even more so by changes made in response to the first round of reviews.

4) Regarding the 11 isolates selected for AMR analysis. It is not clear that these are representative of each of the genotypes associated with the presence of various AMR genes. This should be made explicitly clear.

We thank the reviewer for this suggestion and have specified the selection of the isolates for AMR phenotypic testing in NL343. *“We selected 11 E. albertii (HAC n= 7, BAC n= 4) isolates that captured the lineage and genotypic AMR diversity across the phylogenetic tree”*

5) The authors have completed an in silico LEE analysis and identified that this region is present at a high level of similarity in only a very small number of isolates, and this data is buried in supplemental Figure 2 (and additionally interesting that the LEE and eae homology are not congruent). The presence and function of eae/intimin is suspect without at least some of the T3SS, yet the authors elect not to mention this in the body of the manuscript. Additionally, in any of the isolates that have a complete LEE, is it functional? It is understood that the authors do not wish to complete functional assays on the Shiga toxin, but the a functional LEE would be a significant in a non-E. coli species, especially considering that a functional region is lacking in most isolates.

The data was not intended to be ‘buried’ in Supplementary as the reviewer states – these were additional analyses are of low relevance to the key question of the study (the ecology of *E. albertii*) and were performed in response to the reviewer’s comments on the first version of the manuscript so were added to the Supplementary data. We would be very happy to take editorial advice to move this into the main text if the editor felt that was warranted.

Furthermore, as indicated by the legend, the mapping coverage against the LEE pathogenicity island was $\geq 70\%$ in all isolates, so the presence of the eae gene is not ‘suspect’ as the reviewer suggests. We agree with the reviewer that this work opens up several exciting new avenues of wet laboratory work outside the scope of this study, including functional exploration of the Stx toxin and LEE island variants. To facilitate this, we have elected to explicitly make isolates from this study available for further work from functional microbiology groups (please see contact details See NL 523.